# Combination Treatment Strategies to Overcome PARP Inhibitor Resistance

**DOI:** 10.3390/biom13101480

**Published:** 2023-10-03

**Authors:** Young-Hwa Soung, Jun Chung

**Affiliations:** Department of Pathology, Renaissance School of Medicine, Stony Brook University, Stony Brook, NY 11794, USA; younghwa.song@stonybrookmedicine.edu

**Keywords:** PARP-1, PARP inhibitors, resistance, combination therapy, homologous recombination deficiency, breast cancers, ovarian cancers

## Abstract

Poly(ADP-ribose) polymerase (PARP) enzymes have been shown to be essential for DNA repair pathways, including homologous recombination repair (HRR). Cancers with HRR defects (e.g., BRCA1 and BRCA2 mutations) are targets for PARP inhibitors (PARPis) based on the exploitation of “synthetic lethality”. As a result, PARPis offer a promising treatment option for advanced ovarian and breast cancers with deficiencies in HRR. However, acquired resistance to PARPis has been reported for most tumors, and not all patients with BRCA1/2 mutations respond to PARPis. Therefore, the formulation of effective treatment strategies to overcome resistance to PARPis is urgently necessary. This review summarizes the molecular mechanism of therapeutic action and resistance to PARPis, in addition to emerging combination treatment options involving PARPis.

## 1. Introduction

Dysfunction in DNA repair often leads to genetic instability, which enables tumorigenesis [1]. In addition, malignancy and resistance to treatment are often linked to mutations of tumor suppressor genes that regulate DNA-damaging responses (e.g., BRCA1, BRCA2, PALB2, RAD51C, RAD51D, MLH1) or gatekeeper genes that trigger cell cycle arrest in response to DNA damage (e.g., p53, ATM, CHK1, CHK2) [2]. Although DNA repair dysfunction induces cancer development, it also opens the door for novel therapeutic development because cancers with DNA repair dysfunction tend to depend on alternative DNA repair systems to survive under genotoxic stress [3]. Therefore, it is important to identify cancer-specific vulnerabilities in cancers with DNA repair dysfunction to enhance the therapeutic potential and overcome drug resistance. The concept of “synthetic lethality” arises from a situation in which the impairment of more than two DNA repair components results in cell death, and it emerges as a promising strategy for novel therapies by targeting the DNA repair pathways in cancers [4].

As mentioned above, many cancers display a deficiency in one of the genes that are involved in homologous recombination (HR), which is an essential process to repair DNA double-stranded breakages (DSBs) [3]. The BRCA1 and BRCA2 proteins play important roles in HR by the error-free repair of DSBs [5]. In addition to BRAC1/2, there are several genes, such as PALB2 and RAD51, that contribute to HR [6]. Mutations of these genes are categorized as ‘BRCAness’ phenotypes represented by HR deficiency (HRD) [7]. ‘BRCAness’ phenotypes have served as biomarkers in predicting the therapeutic efficacy of HR inhibitors such as PARP inhibitors (PARPis) in breast, ovarian, pancreatic, and prostate cancers [8,9,10,11]. Currently, the next-generation sequencing (NGS)-based myChoice CDx and FoundationOne CDx assays are to validate the HRD status of cancers [12]. However, it remains challenging to establish a standardized test to understand the complex mechanisms leading to HRD. According to the Cancer Genome Atlas, roughly 50% of high-grade serous ovarian cancers (HGSOC) and 10–20% of triple-negative breast cancers (TNBCs) and metastatic prostate and pancreatic cancers may harbor BRCA1/2 mutations and other aberrations in HRD, qualifying them as potential candidates for treatment with PARPis [8,9,10,11]. However, it is common to observe acquired resistance to PARPis among cancer patients, even those with BRCA mutations [13]. Further, recent studies analyzing larger numbers of TNBC cell lines showed that sensitivity to PARPis is not necessarily related to the BRCA status [14], suggesting that BRCA-independent novel HR pathways may provide resistance to PARPis. Therefore, there is an urgent need to understand the nature of HRD in cancers resistant to PARPis.

## 2. Mechanisms of DNA-Damaging Responses

When genotoxic events produce DNA single-stranded breaks (SSBs) and DNA double-stranded breaks (DSBs), cells respond by activating DNA-damaging responses (DDR). Nucleotide excision repair (NER), base excision repair (BER), and mismatch repair (MMR) are the mechanisms that repair SSBs, whereas homologous recombination (HR) and non-homologous end joining (NHEJ) are the mechanisms that respond to DSBs [15,16]. HR usually occurs during the S and G2 phases of the cell cycle and uses the sister chromatid as a template to replace the DNA sequence and preserve genetic information [17]. On the other hand, NHEJ repairs the broken DNA ends through non-homologous end-joining and occurs throughout the cell cycle [17]. NHEJ is shown to be prone to errors in comparison to the high-fidelity editing of HR. The incorrect repair of DNA damage can lead to mutations and chromosomal aberration, which contribute to tumorigenesis.

During the process of HR, the Mre11-Rad50-Nbs1(MRN) complex initiates HR at the DSB by recruiting TIP60 and ATM to the DNA [18,19]. The phosphorylation of H2AX by TIP60/ATM allows phosphorylated H2AX to serve as an anchor for MDC1 [20]. Then, ATM also phosphorylates MDC1 so that phosphorylated MDC1 recruits E3 ligases RNF8 and RNF168, which ubiquitinate H2AX, which brings p53 binding protein (53BP1) and BRCA1 to the DDR foci [21]. The next step in HR involves the collaboration of nucleases such as EXO1, DNA2, and MUS8 with the MRN complex to resect the DSB ends, which leads to single-stranded DNA (ssDNA) formation [22]. Next, hyperphosphorylated replication protein A (RPA) coats the resected 3′ end, which is then replaced by RAD51 [23]. These events generate nucleoprotein filaments. BRCA2 and PALB2 also favor nucleoprotein filament formation as BRCA2 binds to BRCA1 and promotes the loading of recombinase RAD51 on ssDNA [24]. Once RAD51 is loaded on the ssDNA, it protects the DNA ends from degradation and mediates the invasion of homologous sequences and D-loops [25]. Therefore, we can assume that the restoration of the HR process plays an important role in resistance to PARPis based on their ability to repair DSBs, which otherwise lead to cell death. 

## 3. The Mechanism by which PARP Inhibitors Act

Poly(ADP-ribose) polymerase-1 (PARP-1) is an ADP-ribosylating nuclear enzyme that regulates diverse forms of DNA repair [26]. Among the 17 PARP family members of ADP-ribosyl transferases, PARP1 represents 80% of the poly ADP ribosylation (PAR) activity in cells as a founding member of the PARP family [27]. As PARP1 plays a key role in DNA repair and is reported to be upregulated in various cancer cell lines and tumor tissues, PARP1 has been a major target of PARPis in clinical trials [28]. PARP1 serves as a critical sensor and signal transducer of SSBs and DSBs and binds to exposed DNA. Upon binding to DNA, it undergoes conformational changes to be catalytically active and initiates the DNA repair processes [28]. PARP1 is eventually released from the repaired DNA region thorough self-PARylation. PARPis bind to key NAD+-binding and catalytic residues of PARP1 and inhibit the auto-PARylation of PARP1 at its auto-modification domain, which leads to ‘PARP trapping’ by blocking the release of PARP1 from DNA [29]. The consequence of PARP trapping leads to the crosslinking of DNA proteins, the collapse of replication forks, and a subsequent increase in DSBs [29,30,31] (Figure 1).

Four PARPis (Olaparib, Rucaparib, Niraparib, and Talazoparib) are approved by the FDA for the treatment of advanced ovarian and breast cancers with BRCA1/2 deficiency [32], and three PARPis (Veliparib, Pamiparib, and Fluzoparib) are currently under clinical trials for evaluation [33]. Different PARPis can induce various allosteric changes in PARP1 upon binding, which impacts PARP trapping and the subsequent cytotoxicity [34]. Among these PARPis, the most potent PARP trapping activity is reported in Talazoparib (nearly 100-fold more effective than the rest of the PARPis), followed by Niraparib, Olaparib, and Rucaparib [29,31]. The cytotoxic activity was also found to be highest in Talazoparib among PARPis. However, the relationship between the PARP trapping capability and efficacy of PARPis has not been clarified yet and additional studies are needed to understand the allosteric effects of different PARPis on PARP inhibition.

## 4. Mechanisms of Resistance to PARP Inhibitors

Although PARPis contributed to progression-free survival and overall survival, more than 40% of patients with a BRCA1/2 deficiency fail to respond to PARPis [35,36,37]. Furthermore, the prolonged administration of PARPis often leads to acquired PARPi resistance among patients. Therefore, resistance to PARPis is not uncommon in the clinic, and understanding the mechanisms of PARPi resistance in detail is necessary to increase PARPis’ sensitivity. We summarize the potential mechanisms of inherent and acquired resistance to PARPi therapy (Figure 1).

### 4.1. Restoration of Homologous Recombination Proficiency

The restoration of HR in HR-deficient tumors represents the most common acquired resistance mechanism to PARPis [37,38]. The restoration of HR activity can be achieved directly (direct effects on the HR machinery through genomic, epigenetic, and post-translational alterations) or indirectly (i.e., growth factor receptor-mediated signaling pathways that increase expression or activity of HR machinery) [38]. The direct restoration of HR includes genetic reversion mutations such as somatic insertion or deletion mutations that lead to the expression of functional proteins to increase HR proficiency [36]. For example, a germline or somatic reversion mutation in BRCA1/2, RAD51C, RAD51D, and PALB2 can restore protein functions relevant to DNA damage repair and thus confer up to 25% of resistance to PARPis among patients with BRCA1/2 deficiencies [38,39]. Another example of direct HR restoration is epigenetic regulation involving the promoter methylation of BRCA1 and RAD51C [40]. The loss of *BRCA1* and *RAD51C* promoter methylation (de-methylation) restores the functional expression of these two proteins and leads to resistance to PARPis [40]. Oncogenic signaling pathways including the VEGF receptor, PI3 kinase, and heat-shock protein 90 (HSP90) were shown to promote HR proficiency indirectly by increasing the expression of DDR-associated genes [38]. Therefore, the therapeutic targeting of these pathways in combination with PARPis can be considered to overcome PARPi resistance.

Regaining HR proficiency can be achieved without affecting the BRCA1 mutation status, especially BRCA1 mutant cancers. The most studied example is the inactivation of the TP53BP1 gene that encodes the 53BP1 protein [41]. In BRCA1 mutated breast cancer cells, the loss of 53BP1 rescues the BRCA1 deficiency and prevents genomic instability by activating the ATM-mediated processing of broken DNA ends [41,42]. Then, 53BP1 recruits a protein complex called the Shieldin complex, composed of SHLD1, SHLD2, SHLD3, and REV7, to the DSB sites [43]. A number of recent studies indicate the role of the Shieldin complex as the key regulator of NHEJ repair and HR repair, both of which are associated with PARPi resistance [43,44,45].

### 4.2. Stabilization of Replication Forks

BRCA1 and BRCA2 are required for the protection of stalled replication forks by preventing them from being attacked by nucleases such as MRE11, DNA2, and MUS81 [46,47]. Therefore, cells need to rely on BRCA1 and BRCA2 for their survival when PARPis trap PARP on SSBs and stalled replication forks occur. PARPi-resistant BRCA mutant cells have developed an alternative mechanism to protect DNA replication forks from nucleases that attack the stalled replication forks [48,49]. PARPi-resistant BRCA mutant cells have low levels of EZH2, which leads to reduced H3K27 methylation and prevents the recruitment of nuclease MUS81 at stalled replication forks [49]. In addition, PTIP deficiency inhibits the recruitment of nuclease MRE11 to stalled replication forks and protects nascent DNA from degradation in BRCA2-deficient cells [50]. FANCD2 is highly overexpressed in BRCA1/2 mutant breast cancers and contributes to DNA fork stability by inhibiting MRE11-mediated DNA replication fork degradation in BRCA1/2 mutant breast cancer cells [51,52]. As fork degradation is an important synthetic lethality mechanism of PARPis, the increased stabilization of stalled replication forks by the various mechanisms mentioned above confers resistance to PARPis.

### 4.3. DNA POLQ

DNA polymerase θ, also known as DNA POLQ, is an enzyme that is involved in DSB repair by contributing to the error-prone “microhomology-mediated end joining (MMEJ) pathway” [53,54]. DNA POLQ is highly expressed in HR-deficient ovarian and breast cancers [53,54]. DNA POLQ inhibitors present synthetic lethality with PARPis in BRCA1/2 mutant tumor cells with acquired resistance to PARPis [55,56]. DNA POLQ is able to bind RAD51 and inhibits RAD51-mediated HR. Based on the potential of DNA POLQ inhibitors to overcome acquired resistance to PARPis in HR-deficient tumors, a phase I and II study that combines a DNA POLQ inhibitor, ART4215, with a PARP inhibitor, Talazoparib, was recently initiated in BRCA mutant breast cancer.

### 4.4. Alterations in Cell Cycle Control

The overexpression of cell cycle regulators such as cyclin-dependent kinase 12 (CDK12) and WEE1 is implicated in PARPi resistance by restoring HR [57]. Knocking down CDK12 expression increases the sensitivity to PARPis by downregulating the expression of DNA repair proteins [57]. CDK12 inhibitors reversed de novo and acquired resistance to PARPis in BRCA1 mutant and BRCA wild-type triple negative breast cancer cells [58]. In addition, an inhibitor of WEE1 kinase (WEE1i), Adavosertib, synergized well with PARPis in several preclinical studies by abrogating G2 cell cycle checkpoints. The inhibition of WEE1 activity allows HR-deficient cells to prematurely enter into mitosis, which increases DNA DSBs [59].

### 4.5. Inhibition of PARP Trapping by PAR Glycohydrolase (PARG) Downregulation 

The depletion of poly(ADP-ribose) (PAR) glycohydrolase (PARG) expression by shRNA counteracts PARPi-mediated synthetic lethality by restoring PARP1 signaling and PARylation, even in the presence of PARPis [60]. As mentioned in Section 3, PARP trapping induced by PARPis leads to cell death via the accumulation of unrepaired SSBs and subsequent DSBs [31]. An increase in PARylation due to the loss of PARG expression includes PARP1 auto-PARylation, which allows PARP1 release from SSBs and the inhibition of PARP1 trapping. Consistent with this finding, the depletion of PARG expression is observed in PARPis resistant triple-negative breast cancers and serous ovarian cancers [60], suggesting that the loss of PARG leads to PARPi resistance.

### 4.6. Increased PARPi Efflux by P-gp Efflux Pumps

The chromosomal translocation of drug efflux transporter genes called Abcb1a and Abcb1b has been reported to increase the efflux of PARPis and contribute to PARPi resistance [61]. Abcb1a and Abcb1b encode P-glycoprotein (P-gp) efflux pumps, which are implicated in chemoresistance among various drugs and in the reduction of the efficiency of these compounds [62]. Consistent with these observations, Olaparib-resistant breast cancers showed a 2–85-fold increase in Abcb1a/b expression [62]. The P-glycoprotein (P-gp) efflux pump inhibitor Tariquidar effectively restored the sensitivity of PARPi-resistant ovarian cancer cells to PARPis [63,64]. Two common Abcb1a/b inhibitors, Verapamil and Elacridar, reversed the resistance of Paclitaxel- and Olaparib-resistant ovarian cancer cells [65]. 

### 4.7. ssDNA Gap Suppression

Recent evidence suggests that ssDNA gaps determine genotoxic lesions and PARPi sensitivity [66,67,68]. ssDNA gaps can occur in both the leading and lagging strands. ssDNA was indeed the predominate lesion identified after various chemo-treatments, and SSDNA gaps were considered in the toxicity of genotoxic agents [66]. Quinet et al. demonstrated that the primase-polymerase (PRIMPOL)-mediated repriming of DNA synthesis suppresses fork reversal, indicating a protective role of PRIMOL in BRCA-deficient cells [69]. They further determined that it was the primase activity of PRIMPOL, not the polymerase activity, that inhibited fork degradation in BRCA-deficient cells, thus leaving ssDNA gaps behind the replication forks. Kang et al. showed that PRIMPOL repriming and ssDNA gap accumulation are suppressed by BRCA2 and MCM10 [70]. These studies support the role of PRIMPOL-mediated ssDNA gap formation and the suppression of these gaps as an important mechanism to determine PARPi resistance in BRCA-deficient tumors. 

## 5. The Feasible Combination Treatment Options to Overcome Resistance to PARP Inhibitors

Resistance to PARPis indicates that the combination of PARPis with a variety of existing treatment options may provide a solution for those tumors that fail to reach synthetic lethality with PARPi monotreatment. We summarize the various clinical trials that predict the best combination treatment to maximize the therapeutic efficacy of PARPis (Table 1).

### 5.1. PARPis and Chemotherapy Drugs

PARPis available in the clinic are Olaparib (Lynparza, AsteraZeneca, Cambridge, UK), Rucaparib (Rubraca, Clovis Oncology, Boulder, CO, USA), Niraparib (Zejula, Gsk), and Talazoparib (Talzenna, Pfizer, New York, NY, USA). PARPi combination therapy with chemotherapy drugs (Cyclophosphamide, Carboplatin, and Paclitaxel) has been studied and used by many clinicians and researchers to treat patients with platinum-sensitive recurrent and advanced ovarian and breast cancers [32,71]. Among these PARPis, Olaparib effectively enhanced the efficacy of DNA-damaging chemotherapy drugs such as Carboplatin and Cisplatin [72,95]. In a randomized phase II trial involving recurrent platinum-sensitive ovarian cancer patients (NCT01081951), the addition of Olaparib (200 mg capsule) to Carboplatin (AUC 4 mg/mL per min) plus Paclitaxel (175 mg/(m^2^)) showed a significant improvement in progression-free survival (PFS) compared to Carboplatin plus Paclitaxel alone [73]. The PFS benefit of the combination of Olaparib with Carboplatin allowed the use of lower doses (Carboplatin AUC 4 mg/mL per min) than those used with chemotherapy alone (Carboplatin AUC 6 mg/mL per min), suggesting that Olaparib enhances the cytotoxic effects of Carboplatin [73]. In particular, ovarian cancer patients with BRCA mutations had the greatest clinical benefits [73]. In another example, Li C et al. compared and evaluated the clinical efficacy and safety of a combination treatment involving PARPis (Veliparib) versus chemotherapy (Carboplatin, Paclitaxel, Placebo, Cyclophosphamide) alone in patients with triple-negative breast cancer by using a meta-analysis through searches of seven databases [74]. The group treated with PARPis plus chemotherapy showed better PFS and overall survival (OS) than the group treated with chemotherapy alone. The studies demonstrated that the PARPis showed a synergy with chemotherapy in TNBC patients regardless of the BRCA mutational status [74]. However, in terms of safety, some specific adverse effects (AEs), including neutropenia, anemia, and diarrhea, were observed in relatively higher numbers in the combination group. The selection of the optimal dose of PARPis and chemotherapy in the combination therapy to maximize the clinical benefits and minimize AEs remains a challenge.

### 5.2. PARPis and Immunotherapy

Given the interaction between DNA damage in tumors and the activated immune system, the combination of PARPis with immune checkpoint inhibitors (ICI) has been considered as the most effective strategy to overcome PARPi resistance. The rationale of this combination is based on two main premises. Firstly, HRD cancers have a high tumor mutational burden (TMB) caused by the deficient DNA repair pathway, which elevates tumor-specific neoantigen loads and activates the antitumor immune response [75,96,97]. Secondly, PARPis suppress anticancer immunity through PD-L1 upregulation. PARPis also promote the accumulation of cytosolic DNA fragments, which activates the DNA-sensing cGAS-STING pathway and thereby induces a type 1 interferon (IFN)-mediated antitumor immune response with T cell recruitment [76,77].

The synergistic effect between PARPis and ICI has been confirmed in the clinical setting, such as the phase I/II TOPACIO (NCT02657889) with Niraparib and Pembrolizumab (PD-1 inhibitor), in platinum-resistant ovarian cancers and metastatic TNBCs [78,79]. The combination of Niraparib (200 mg) and Pembrolizumab (200 mg) showed improved clinical efficacy (ORR of 18% and disease control rate (DCR) of 65%) compared with monotherapy, particularly in patients with BRCA wild-type and non-HRD ovarian cancer [78]. The putative combination of Niraparib and Pembrolizumab was effective in ovarian cancer regardless of BRCA and HRD status, whereas its therapeutic synergistic effect was observed only in the breast cancer cohort with BRCA-mutated tumors among metastatic TNBCs (ORR of BRCAm vs. BRCAwt, 47% vs. 11% and PFS of BRCAm vs. BRCAwt, 8.3 months vs. 2.1 months) [79]. Another clinical trial in the platinum resistance setting is the phase I/II MEDIOLA (NCT02734004), which explored the combination of Olaparib and Durvalumab (PD-L1 inhibitor). Patients received 300 mg of Olaparib twice daily for 4 weeks, followed by a combination of Olaparib (300 mg) and Durvalumab (1.5 g) until tumor progression was observed [80]. In patients with gBRCAm platinum-sensitive relapsed ovarian cancer, the effect of combination therapy showed an ORR of 63% and a 12-week DCR of 81%. In patients with BRCAm HER2-negative metastatic breast cancer, the combination was tolerable, with an ORR of 63.3%, a 12-week DCR of 80%, and a 28-week DCR of 50%. The combination therapy had more effective efficacy compared to the monotherapy in gBRCAm. Recently, this clinical trial was further extensively studied with the BRCA wt cases. Triplet therapy with Olaparib (300 mg), Durvalumab (1.5 g), and Bevacizumab (10 mg/kg) showed enhanced tumor responses (ORR of 77.4%), compared with the doublet therapy (ORR of 31.3%) and PARP or VEGF inhibitor monotherapy [98]. In the OPEB-01 trial (NCT04361370), the synergistic effect and safety of the triplet combination was also evaluated as a maintenance treatment for BRCA wt patients with platinum-sensitive recurrent ovarian cancer [99].

Besides this, several larger randomized phase III trials with a combination of PARPis and ICI are currently being performed in the first-line setting and in pretreated patients. For example, the phase III ENGOT-OV44/FIRST trial (NCT03602859) is being performed to compare the clinical efficacy, such as PFS and OS, and safety of Niraparib (as a maintenance drug) + chemotherapy (Paclitaxel and Carboplatin or Bevacizumab) + Dostarlimab (PD-1-blocking antibody) vs. Niraparib (as a maintenance drug) + chemotherapy in patients with FIGO stage 3 or 4 non-mucinous epithelial ovarian cancer [81]. ENGOT-OV43/KEYLYNK-001 (NCT03740165) is currently evaluating the use of Olaparib as a first-line maintenance therapy after Pembrolizumab (PD-1 inhibitor) plus chemotherapy in BRCA1/2-nonmutated advanced epithelial ovarian cancer [82]. With a similar premise, the phase III ATHENA trial (NCT03522246) enrolling 1000 patients is evaluating the efficacy of the combination of Rucaparib with Nivolumab (PD-1-blocking antibody) as a maintenance therapy in newly diagnosed high-grade epithelial ovarian cancer patients. In this study, ATHENA COMBO aims to determine whether the addition of Nivolumab to Rucaparib increases the activity of Rucaparib and extends PFS compared to Rucaparib alone following standard platinum-based chemotherapy [83]. In addition, biomarkers such as PD-L1 expression and HR deficiency status, related to the prediction of PARPi resistance and response, were assessed in this combination study. Most recently, the results of the phase III DUO-O trial (NCT03737643) were presented at the 2023 ASCO Annual Meeting. This trial included 1130 patients and was designed to assess the efficacy and safety of triple treatment with Bevacizumab plus Olaparib and Durvalumab as a maintenance therapy in newly diagnosed advanced ovarian cancer patients without BRCA mutations [86]. They reported that the triplet maintenance therapy provided a statically significant and clinically meaningful improvement in PFS compared to Bevacizumab monotherapy in patients with HRD-positive tumors (median PFS of triplet arm, 37.3 months; median PFS of Bevacizumab alone arm, 23 months). In the intent-to-treat (ITT) population, the triplet arm showed a median PFS of 24.2 months vs. 19.3 months in the Bevacizumab alone arm (19.3 months). The final analysis including OS and other secondary endpoints is still awaiting a definitive conclusion. Numerous other clinical trials are now underway and are investigating the safety, toxicity, and optimal dose of combination PARPi and ICI agents in the target population (NCT02849496) [84].

### 5.3. PARPis and Anti-Angiogenic Agents

Preclinical studies have demonstrated that anti-angiogenic agents can increase PARPi susceptibility through HR deficiency (HRD) by downregulating homologous recombination repair genes such as BRCA1 and RAD51 under hypoxia stress [85]. This suggests that the combination of anti-angiogenic agents and PARPis may have at least additive or synergic effects in PARPi-resistant cancers. Cediranib, as an oral anti-angiogenic agent, is a VEGF receptor-targeted agent. In a randomized phase II trial (NCT01116648), the addition of Cediranib (30 mg) to Olaparib (200 mg) improved the PFS (from 9 months to 17.7 months) compared with Olaparib monotherapy (400 mg) in a population of women with platinum-sensitive high-grade or endometrioid ovarian cancer [100]. The survival benefit of the PARPi combination therapy with Cediranib was observed in both BRCA mutant and BRCA wild-type populations [101]. A subsequent study of Olaparib and Cediranib combination treatment in platinum-resistant ovarian cancer patients without BRCA1/2 mutation is currently underway in clinical trials (NRG-GY005:NCT02502266) [87]. In another clinical trial, Olaparib and Bevacizumab combination treatment effectively improved PFS compared to a placebo plus Bevacizumab in HRD-positive advanced ovarian cancer patients who received first-line platinum-based chemotherapy (PAOLA-1/ENGOT-ov25 trial; NCT02477644) [102]. In other study, patient cohorts that received Olaparib (300 mg for 24 months) plus Bevacizumab (15 kg for 15 months) showed significant improvements in PFS2 (second progression or death) and TSST (second subsequent therapy or death) [103]. A subsequent PAOLA-1 study recently updated the clinically meaningful OS efficacy in HRD-positive ovarian cancer patients who received Olaparib + Bevacizumab. This study showed the potential of Olaparib + Bevacizumab as one of the standard treatments for HRD+ newly diagnosed advanced ovarian cancer patients [104]. While the combination of Olaparib and Bevacizumab is approved only for HRD-positive cancers, the combination of Niraparib plus Bevacizumab has been reported to have clinical efficacy regardless of HRD status. The combination of Niraparib plus Bevacizumab is more effective than Niraparib alone in platinum-sensitive recurrent ovarian cancer (NTC02354131) [105].

### 5.4. PARPis and PI3K/AKT Pathway Inhibitors

The PI3K/AKT/mTOR pathway is known to contribute to DSB repair through HR. PI3K inhibitors induce HRD by downregulating BRCA1/2. mTOR inhibitors suppress DSB repair through decreasing homologous recombination repair (HRR) gene expression [88,106]. The synergistic effect has been confirmed in a phase I clinical trial with Olaparib and Alpelisib (FDA-approved α-specific PI3K inhibitor) in combination in BRCA wild-type platinum-resistant ovarian cancer and breast cancers [89,107]. The maximum tolerated dosage when combining Alpelisib (200mg once daily) and Olaparib (200 mg twice daily) was determined in a phase Ib dose-escalation/dose-expansion clinical trial (NCT01623349). The overall response rate (ORR) of the Olaparib and Alpelisib combination (33%) was higher than that of the Alpelisib (<5%) or Olaparib (4–5%) monotherapy in patients with platinum-resistant epithelial ovarian cancer [89]. A similar clinical study, EPIKK-O/ENGOT-ov61 (NCT04729387), is ongoing to assess the efficacy and safety of Olaparib + Alpelisib vs. single-agent chemotherapy in patients with platinum-resistant or refractory high-grade serious epithelial ovarian cancer (HGSOC) with BRCA wild type [108]. A recent study reported clinical efficacy with the combination treatment of Niraparib and Copanlisib (PI3K inhibitor) in recurrent high-grade serous or BRCA mutant ovarian cancer (NCT03586661) [109]. This treatment is now being evaluated for optimal doses and side effects. Several other ongoing clinical trials of PARPis in combination with mTOR inhibitors (Vistusertib;AZD2014) are also evaluating the clinical efficacy, safety, and best dose of Olaparib and Vistusertib in triple-negative breast cancer (NCT02208375) [110]. The ComPAKT trial evaluated the synergy in the combination of a pan-AKT inhibitor, Capivasertib (AZD5363), and Olaparib (NCT02338622) [111]. A 44.6% clinical benefit (RECIST complete response/partial response or stable disease ≥ 4 months) was achieved in platinum-resistant HGSOC patients regardless of BRCA or DDR status and PI3K-AKT pathway alterations [111].

### 5.5. PARPis and RAS/RAF/MEK Pathway Inhibitors

PARPi resistance is related to the upregulation of the RAS/MAPK pathway, suggesting that the MAPK pathway may have a certain influence on the re-sensitization to PARPis. MEKi decreases the HR capacity and causes increased DNA damage accumulation [112,113]. The combination of MEKi and PARPis served to induce more DNA damage and apoptosis in RAS mutant cell lines or BRCA2 proficient cancers [112,113]. The clinical benefits of this combination in patients are currently under investigation in an ongoing phase I/II trial of Olaparib and Selumetinib (MEKi), which includes a cohort with PARPi-resistant ovarian cancer (NCT03162627) [114].

### 5.6. PARPis and Epigenetic Drugs

DDR genes are epigenetically modified to cause transcriptional silencing and the loss of DNA repair capacity in cancers. These epigenetic modifications have been reported to affect PARPis’ sensitivity [115,116]. Therefore, epigenetic inhibitors including the DNA methyltransferase inhibitor (DMNTi) and histone deacetylation inhibitor (HDACi) have been considered to enhance PARPi treatment outcomes by modifying epigenetic marks associated with resistance. The combination of PARPis and epigenetic drugs has been tested in preclinical and clinical settings [117,118]. The addition of Decitabine (DMNTi) to Talazoparib enhanced the efficacy of Talazoparib by inducing the tight binding of PARP1 and DNMT to chromatin in BRCAness acute myeloid leukemia (AML) [119]. In a phase I clinical trial, the dose of the combination of Decitabine (20 mg/m^2^ intravenously daily for 5 or 10 days) and Talazoparib (1 mg orally daily for 28 days) was evaluated, based on tolerability, efficacy, and pharmacodynamic data (NCT02878785) [120]. At present, the NCT03742245 trial is ongoing to determine the safety and efficacy of the combinational treatment of Olaparib and Vorinostat (HDACi) in metastatic breast cancer [121].

### 5.7. PARPis and ATR/CHK1 Inhibitors

One of the strategies in modulating DNA repair activity in HR-deficient tumors is to disrupt cell cycle checkpoint signaling. Under elevated DNA replication stress and DNA-damaging conditions, activated ataxia telangiectasia and rad3-related (ATR) and its downstream checkpoint kinase 1 (CHK1) stabilize replication forks and simultaneously activate the S and G2-M checkpoints, resulting in DNA repair [122,123]. The ATR-CHK1 pathway also prevents collapse into DNA DSBs [122,123]. Thus, inhibition of the ATR-CHK1 pathway collapses replication forks and disrupts cell cycle progression. These events lead to chromosome aberrations, mitotic catastrophe, and ultimately apoptosis. Given that the HR-independent mechanism of PARPi resistance is associated with the increased stabilization of replication forks, a combination strategy targeting the ATR-CHK1 pathway is expected to be a rational approach to overcome PARPi resistance. Preclinical studies have demonstrated the synergistic effect of a PARPi + ATRi combination in in vitro and in vivo, where ATRi sensitized BRCA1/2-deficient ovarian cancer cells and tumors to PARPis by increasing DNA replication fork instability, double-strand breaks, and apoptosis [124,125]. Fonia et al. reported that Olaparib plus Ceralasertib (ATRi) was well-tolerated in platinum-resistant high-grade serous epithelial ovarian cancer (HGSOC) patients. The phase II study of Olaparib plus Ceralasertib (CAPRI: NCT03462342) showed their synergy in BRCA1 mutant ovarian cancers [90]. Moreover, their recent clinical study determined the tolerability, response rate, and safety of the combination treatment of Ceralasertib and Olaparib in three different cohorts (platinum-sensitive, platinum-resistant, and platinum-sensitive/PARPi-resistant) [126]. A great benefit from the addition of Ceralasertib was observed in HR-deficient platinum-sensitive/PARPi-resistant HGSOC, increasing the median PFS by 7.43 months (95% CI, 4.73–15.1), with a 57% partial response (PR: *n* = 4/7) and 50% ORR (*n* = 6/12) [126]. Ceralasertib re-sensitized PARPi-resistant HR-deficient ovarian cancer to Olaparib. This combination had dose reductions and low-grade toxicity. Other studies including the combination of Niraparib and Elimsertib (ATRi: BAY 1895344) in recurrent advanced solid tumors or ovarian cancer are underway (NCT04267939) [91]. CHK1 inhibitors (CHK1i) have also shown a benefit in overcoming PARPi resistance. A phase I clinical trial combining Prexasertib (CHK1i) and Olaparib has been completed in BRCA mutant PARPi-resistant HGSOC (NCT03057145) [92]. The addition of Prexasertib suppressed DNA repair via compromised HR and enhanced DNA damage [127]. Future studies are needed to assess the optimized dose for combination to minimize hematologic toxicity.

### 5.8. PARPis and Other DDR Inhibitors

Besides ATR-targeting drugs, the WEE1 inhibitor (WEE1i) is another modulator of DDR that increases replication stress and eventually promotes mitotic catastrophe by controlling CDK activity [128]. A number of preclinical studies have already shown that WEE1i may serve as a combination partner with PARPis. Indeed, the combination of PARPis and WEE1i has been demonstrated to be synergistic in small cell lung cancer PDX, ovarian cancer PDX, and breast cancer models [59,129,130]. The phase II EFFORT clinical study (NCT03579316) reported the efficacy of the combination of Olaparib and Adavosertib (WEE1i) in PARPi-resistant ovarian cancer. The combination therapy achieved clinical benefits compared to Olaparib monotherapy, with ORR (29% vs. 23%), clinical benefit (CBR: 89% vs. 63%), and median PFS (6.8 months vs. 5.5 months). However, due to the high grade 3 or 4 toxicity of this combination, additional studies including dose interruptions and reductions are underway [93].

POLQ inhibitors, known as potent inhibitors targeting tumor-specific DDR, have recently attracted attention as potential combination partners to overcome PARPi resistance. A phase I/II study (NCT04991480) is evaluating the safety and tolerability of a combination with Talzenna (Talazoparib) and ART4215 (selective small-molecule inhibitor of POLQ) in BRCA-deficient breast cancer [94].

## 6. Conclusions

FDA-approved PARPis have served as first-line and second-line maintenance and treatment options for breast, pancreatic, prostate, and ovarian cancers and benefited patients with these cancers with HRD. The recent data suggest the clinical benefits of PARPis in tumors regardless of *BRCA1/2* mutation. However, prolonged treatment with PARPis highlights the emerging issue of PARPi resistance.

In this review, we discussed the restoration of homologous recombination proficiency, the stabilization of replication forks, DNA POLQ, increased drug efflux, PARG mutations, and cell cycle alteration as some of the known mechanisms that contribute to resistance to PARPis. Preclinical studies provided evidence to support these mechanisms, but clinical trials indicated that a combination of multiple mechanisms may be involved in the resistance to PARPis. Therefore, the optimization of combination treatment selection and dosing will be the key to overcome resistance to PARPis in multiple cancer types. We discussed a number of ongoing clinical trials evaluating the rationale of combination strategies to evade PARPi resistance. While the clinical benefits are obvious based on preliminary analyses of these trials, the establishment of the toxicity profiles and the development of reliable predictive biomarkers of the treatment responses will further enhance the use of PARPis.

## Figures and Tables

**Figure 1 biomolecules-13-01480-f001:**
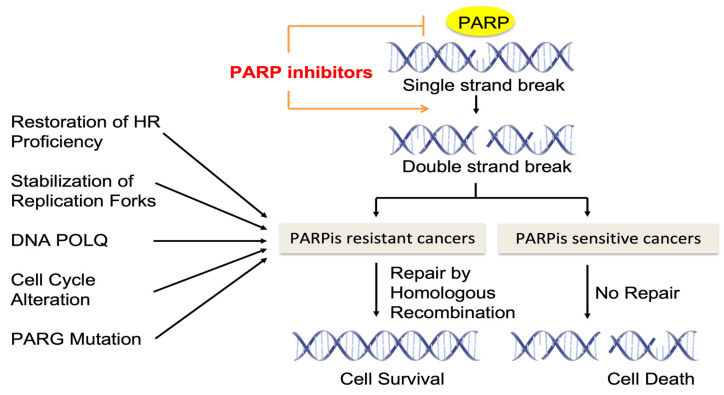
Schematic of PARP inhibitor action and its resistance.

**Table 1 biomolecules-13-01480-t001:** A summary of the main current PARPi combination clinical trials.

	Trial Name	Combination Therapy	Phase	Cohort	Start Completion Date	Ref
PARPi + Chemotherapy	(STUDY41) NCT01081951	Olaparib + Paclitaxel + Carbopatin	II	Platinum sensitive advanced ovarian cancer	4 Feb 2010– 29 Dec 2023	[59]

PARPi + Immunotherapy	(TOPACIO) NCT02657889	Niraparib + Pembrolizumab	II	Platinum-resistance ovarian cancer (regardless BRCA or HRD status) or BRCA mutated metastatic TNBC	15 Apr 2016– 17 Sep 2021	[66,67]

	(ENGOT-OV44) NCT03602859	Niraparib + Dostarlimab	III	Stage III or IV nonmucinous Epithelial ovarin cancer	11 Oct 2018– 22 Jun 2026	[71]

	(MEDIOLA) NCT02734004	Olaparib + Durvalumab	II	gBRCAm platinum-sensitive ovarian cancer	28 Oct 2018– 2026	[68]

	(OPEB-1) NCT04361370	Olaparib + Durvalumab + Bevacizumab	II	Platinum-sensitive recurrent BRCA wt ovarian cancer	28 Oct 2020– Aug 2026	[70]

	(ATHENA) NCT03522246	Rucaparib + Nivolumab	III	Newly diagonsed advanced (FIGO stage III-IV) epithelial ovarin cancer. Fallopian Tube diseases.	14 May 2018– 30 Dec2030	[72]

	(DUO-O) NCT03737643	Olaparib + Durvalumab	III	Newly diagonsed advanced ovarain cancer (Regardless BRCA status)	4 Jan 2019– 25 May 2028	[73]

	NCT02849496	Olaparib + Atezolizumb (Anti-PD-L1)	II	Locally advanced unresectable or metastatic non-HER2 positive breast cancer	30 Mar 2017– 31 Aug 2023	[74]

PARPi + Anti-angiogenic agents	NCT01116648	Olaparib + Cediranib	I/II	Relapsed Platinum sensitive high-grade or endometrioid ovarian cancer	25 Mar 2010– 31 Oct 2018	[75]

	(NRG-GY005) NCT02502266	Olaparib + Cediranib	II/III	Recurrent Platinum-resistant ovarian cancer, Fallopian Tube diseases.	3 May 2016– 30 Jun 2024	[76]

	(PAOLA-1) NCT02477644	Olaparib + Bevacizumab	III	HRD positive high-grade ovarian cancer	6 May 2015– 22 Mar 2022	[77,78,79]
	NCT02354131	Niraparib + Bevacizumab	I/II	platinum-sensitive ovarian cancer	15 Feb 2015– 15 Dec 2021	[80]
PARPi + PI3K/AKT pathway inhibitors	NCT01623349	Olaparib + Alpelisib	I	Platinum-resistant ovarian cancer or recurrent triple negative breast cancer (regardless BRCA or HRD status)	Sep 2012– Dec 2020	[81,82]

	(EPIK-O) NCT04729387	Olaparib + Alpelisib	III	Platinum-resistant or refractory high-grade serious epithelial ovarian cancer (HGSOC) with BRCA wt.	2 Jul 2021– 29 Jul 2025	[83]

	NCT02208375	Olaparib + Vistusertib or Olaparib + Capivasertib	I/II	Recurrent endometrial, triple negative breast cancer or ovarian cancer	11 Nov 2014– 20 Jun 2024	[84]

	NCT02338622	Olaparib + Capivasertib	I	Platinum-resistant HGSOC (regardless BRCA or HRD status)	31 Mar 2014– 21 Mar 2017	[85]

	NCT03586661	Niraparib + Copanlisib	I	Recurrent high-grade serous or BRCA mutant ovarian cancer	29 Apr 2019– 31 Dec 23	[86]

PARPi + MAPK pathway inhibitors	NCT03162627	Olaprib + Selumetinib	I/II	PARPi resistant ovarian cancer or Solid tumor with RAS pathway alteration	4 Aug 2017– 30 Aug 2026	[87]

PARPi + Epigenetic drugs	NCT02878785	Talazoparib + Decitabine	I/II	Acute Myeloid Leukemia	Aug 2016– 19 Nov 2020	[88]

	NCT03742245	Olaprib + Vorinostat	I	Relapsed metastatic breast cancer	11 Jun 2019– 1 Sep 2024	[89]
PARPi + DDR inhibitors	(CAPRI) NCT03462342	Olaparib + Ceralasertib	II	Platinum-resistant HGSOC	9 Mar 2018– 31 Dec 2023	[90]

	NCT04267939	Niraparib + Elimsertib	I	Recurrent advanced solid tumors and ovarian cancer.	26 Feb 2020– 3 Mar 2025	[91]

	NCT03057145	Olaparib + Prexasertib	I	BRCA mutant PARPi resistant HGSOC	10 Mar 2017– 9 Jun 2021	[92]

	(EFFORT) NCT03579316	Olaparib + Adavosertib	II	PARPis resistant ovarian cancer	7 Dec 2018– 30 Dec 2023	[93]

	NCT04991480	Talozoparib + ART4215	I/II	BRCA deficient breast cancer	13 Sep 2021– Aug 2025	[94]

## Data Availability

Not applicable.

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
