# Peer review of "Combination Treatment Strategies to Overcome PARP Inhibitor Resistance"

_biomolecules, 2023, doi:10.3390/biom13101480_

Round 1

Reviewer 1 Report

In the manuscript entitled “Combination treatment strategies to overcome PARP inhibitor resistance”, Soung and Chung provided a synopsis of the molecular mechanism underlying the therapeutic effects of PARP inhibitors (PARPis) along with insights into the resistance mechanisms. Furthermore, the review highlights the existing literature and clinical trials regarding the combined use of PARPi therapies, which could shed light on potential future studies and provide valuable information.

This paper is well written and includes the latest updates. The opinions will undoubtedly be of importance to the field. I have listed below a series of suggestions which would further improve the quality and impact of the manuscript. If addressed, I have no hesitation to suggest publication of this paper.

1. The authors should be more careful about the spellings and languages. The problems include, but are not limited to, line 61 “mechanism” should be “mechanisms”, line 151 “occurs” should be “occur”, line 289 “intent -to-treat” should be “intent-to-treat”. Please pay more attention to the details.

2. Similar as other therapies, the efflux of PARPi through MDRs could lead to PARPi resistance. Please briefly mention this in part 4.

3. Recently, new concepts of gap filling by HR or gap suppression on leading and/or lagging strands have been linked to PARPi resistance. Please also consider adding related papers in part 4.

4. In the part 4.1, the authors also didn’t mention those indirect ways to restore HR, such as the decrease of 53BP1 and SHLD complex, or increase of end resection in restoring HR.

5. In part 5, the authors didn’t mention another aspect of combination therapy between PARPi and drugs that increase the replication stress, such as ATRi and CHK1i. There could be some clinical trials about such therapy strategy.

Author Response

We would like to thank two reviewers who dedicate their time for this review articles and suggested the constructive comments that help us to revise the manuscript. The changes made are highlighted below.

#. Reviewer 1 ,Comments and Suggestions for Authors

In the manuscript entitled “Combination treatment strategies to overcome PARP inhibitor resistance”, Soung and Chung provided a synopsis of the molecular mechanism underlying the therapeutic effects of PARP inhibitors (PARPis) along with insights into the resistance mechanisms. Furthermore, the review highlights the existing literature and clinical trials regarding the combined use of PARPi therapies, which could shed light on potential future studies and provide valuable information.

This paper is well written and includes the latest updates. The opinions will undoubtedly be of importance to the field. I have listed below a series of suggestions which would further improve the quality and impact of the manuscript. If addressed, I have no hesitation to suggest publication of this paper.

  1. The authors should be more careful about the spellings and languages. The problems include, but are not limited to, line 61 “mechanism” should be “mechanisms”, line 151 “occurs” should be “occur”, line 289 “intent -to-treat” should be “intent-to-treat”. Please pay more attention to the details: We corrected these minor spelling errors throughout the manuscript.

  1. Similar as other therapies, the efflux of PARPi through MDRs could lead to PARPi resistance. Please briefly mention this in part 4: We added efflux of PARPis as one of resistance mechanisms in section 4.6.

  1. Recently, new concepts of gap filling by HR or gap suppression on leading and/or lagging strands have been linked to PARPi resistance. Please also consider adding related papers in part 4.: We added the content of ssDNA gap suppression as a mechanism to PARPis resistance in section 4.7.

  1. In the part 4.1, the authors also didn’t mention those indirect ways to restore HR, such as the decrease of 53BP1 and SHLD complex, or increase of end resection in restoring HR.: We added the content regarding 53BP1 and SHLC complex in section 4.1.

  1. In part 5, the authors didn’t mention another aspect of combination therapy between PARPi and drugs that increase the replication stress, such as ATRi and CHK1i. There could be some clinical trials about such therapy strategy.: We added these contents in section 5.7 in the revised version.

Reviewer 2 Report

Although there is no great novelty in terms of information and ideas discussed, this review article provides a good overview of some of the recent PARPi combination trials and follow-on trials. However, much of this has been reviewed recently, and more comprehensively (e.g. https://www.nature.com/articles/s41416-023-02326-7, which has a very similar structure to this manuscript). Therefore I am not sure this review is really covering new ground.  

None of the treatment strategies discussed are really a mechanism-based approach to overcoming PARPi resistance; they are simply trials that are recruiting PARPi resistant patients, mostly for practical reasons.

The section on PARPi resistance is incomplete. Mutation of TP53BP1/Shieldin pathway and ABCB1 transporter fusions should be discussed and studies cited, as these have actually been observed in patients with acquired resistance. Many of the resistance mechanisms cited are only weakly supported by clinical evidence, if at all.

Line 217: It should be mentioned that iniparib is not a PARP inhibitor (or don't mention it at all) - this risks confusing the literature.

Combinations with ATR inhibitors and other DDR drugs are also missing from this review.

Generally good, minor proofreading required (e.g. PAPRi instead of PARPi).

Author Response

We would like to thank two reviewers who dedicate their time for this review articles and suggested the constructive comments that help us to revise the manuscript. The changes made are highlighted below.

#. Reviewer 2,Comments and Suggestions for Authors

Although there is no great novelty in terms of information and ideas discussed, this review article provides a good overview of some of the recent PARPi combination trials and follow-on trials. However, much of this has been reviewed recently, and more comprehensively (e.g.https://www.nature.com/articles/s41416-023-02326-7, which has a very similar structure to this manuscript). Therefore I am not sure this review is really covering new ground. : We added the novel and recent findings in PARPis research and clinical trials in section 4.7, 5.7 and 5.8.

None of the treatment strategies discussed are really a mechanism-based approach to overcoming PARPi resistance; they are simply trials that are recruiting PARPi resistant patients, mostly for practical reasons.: We appreciate this comment from reviewer 2, and certainly agree that the current trials are being performed in need-based rather than the mechanism-based. In this review, we would like to provide the balanced contents between the mechanism-based (section 1-4) and the up to date trial outcome-based (section 5) outcomes. We acknowledge that these two contents do not necessarily correlate.

The section on PARPi resistance is incomplete. Mutation of TP53BP1/Shieldin pathway and ABCB1 transporter fusions should be discussed and studies cited, as these have actually been observed in patients with acquired resistance. Many of the resistance mechanisms cited are only weakly supported by clinical evidence, if at all.: Mutation of TP53BP1/Shieldin pathway and ABCB1 transporter fusions are added in section 4.1 and 4.6 in the revised version.

Line 217: It should be mentioned that iniparib is not a PARP inhibitor (or don't mention it at all) - this risks confusing the literature.: We removed iniparib part to avoid the confusion in the revised version.

Combinations with ATR inhibitors and other DDR drugs are also missing from this review.: We added this content in section 5.7 and 5.8 in the revised version.

Comments on the Quality of English Language: Generally good, minor proofreading required (e.g. PAPRi instead of PARPi).: Minor spelling errors have been fixed in the revised version.